# A bimodal distribution of haze in Pluto's atmosphere

Siteng Fan [1,2✉], Peter Gao[3], Xi Zhang[4], Danica J. Adams[1], Nicholas W. Kutsop[5], Carver J. Bierson[4,6], Chao Liu[1,7], Jiani Yang[1], Leslie A. Young [8], Andrew F. Cheng[9] & Yuk L. Yung[1,10]

Pluto, Titan, and Triton make up a unique class of solar system bodies, with icy surfaces and chemically reducing atmospheres rich in organic photochemistry and haze formation. Hazes play important roles in these atmospheres, with physical and chemical processes highly dependent on particle sizes, but the haze size distribution in reducing atmospheres is currently poorly understood. Here we report observational evidence that Pluto's haze particles are bimodally distributed, which successfully reproduces the full phase scattering observations from New Horizons. Combined with previous simulations of Titan's haze, this result suggests that haze particles in reducing atmospheres undergo rapid shape change near pressure levels ~0.5 Pa and favors a photochemical rather than a dynamical origin for the formation of Titan's detached haze. It also demonstrates that both oxidizing and reducing atmospheres can produce multi-modal hazes, and encourages reanalysis of observations of hazes on Titan and Triton.

[1] Division of Geological and Planetary Sciences, California Institute of Technology, Pasadena, CA 91125, USA. [2] LMD/IPSL, Sorbonne Université, PSL Research University, École Normale Supérieure, École Polytechnique, CNRS, Paris 75005, France. [3] Earth and Planets Laboratory, Carnegie Institution for Science, Washington, DC 20015, USA. [4] Department of Earth and Planetary Sciences, University of California Santa Cruz, Santa Cruz, CA 95064, USA. [5] Astronomy Department, Cornell University, Ithaca, NY 14853, USA. [6] School of Earth and Space Exploration, Arizona State University, Tempe, AZ 85281, USA. [7] Key Laboratory for Aerosol-Cloud-Precipitation of China Meteorological Administration, School of Atmospheric Physics, Nanjing University of Information Science & Technology, Nanjing 210044, China. [8] Southwest Research Institute, Boulder, CO 80302, USA. [9] Johns Hopkins University Applied Physics Laboratory, Laurel, MD 20723, USA. [10] Jet Propulsion Laboratory, California Institute of Technology, Pasadena, CA 91109, USA. ✉email: stfan@gps.caltech.edu

Hazes are common in the atmospheres of solar system bodies and exoplanets[1–4]. They play important roles in atmospheric composition, dynamics, and radiative transfer, most of which are highly dependent on particle sizes. Size distributions of haze particles are tracers of their formation pathways. Multi-modal distributions are indications of multiple formation mechanisms. Sizes of the haze particles in chemically oxidizing atmospheres are usually bimodally/multimodally distributed, as observed in the atmospheres of Earth and Venus[1,2,5–7], whose origins are explained by multiple chemical sources and atmospheric dynamics[1,8]. In contrast, the modality of haze particle distributions in reducing atmospheres is currently unknown. Observations of such hazes are usually interpreted using unimodal haze models for simplicity[9–11], which then encourages simplified physical haze models generating unimodal hazes as well[12,13]. Bimodal size distributions were proposed for transient states of haze formation in some numerical simulations[14], but they have yet to be observed. More constraining observations are therefore required to address such potential oversimplification.

Pluto's atmosphere is comparable with those of Titan and Triton in terms of composition, with $N_2$ being the dominant component, percent-level abundances of $CH_4$, and smaller amounts of $CO$[15–17]. The surface atmospheric pressure of Pluto is similar to that of Triton and equivalent to Titan's atmosphere above 400 km[17–19]. Therefore, despite some differences in temperature and chemical species abundances, haze formation pathways are expected to be similar at this pressure level on these celestial bodies. Investigation of physical processes on one of these worlds thus has significant implications for the others, as well as for chemically reducing atmospheres in general.

The existence of haze in Pluto's atmosphere was confirmed by the New Horizons spacecraft during its flyby in July 2015[11,15,20,21], as well as a recent occultation[22]. Originating from photolysis of $CH_4$ and $N_2$ in Pluto's upper atmosphere driven by solar ultraviolet radiation, haze particles grow through coagulation as they sediment downwards[12,23], which is similar to processes in Titan's upper atmosphere and the atmosphere of Triton[13,24]. Due to the unexpectedly low atmospheric temperature, however, condensation and/or sticking of gaseous species have considerable impacts[23,25,26], which also resemble that in Titan's lower atmosphere and on Triton[13,27]. The haze controls Pluto's energy budget through radiative heating and cooling of the atmosphere and by altering the surface color[28,29]. Understanding the morphology of the haze particles, a tracer of the formation pathways, can determine their interactions with condensing/sticking gases and their role in atmospheric radiation, which is critical for understanding haze in reducing atmospheres.

In this work, we report observations of a bimodal distribution of haze particles in Pluto's chemically reducing atmosphere, which supports the scenario in which single-source organic photochemistry and microphysics can result in multimodal haze particles in chemically reducing atmospheres without contributions from dynamics.

## Results

**Observation and data analysis**. Observations of Pluto's haze have been obtained by multiple instruments onboard the New Horizons spacecraft, with wavelength coverage from the ultraviolet (UV) to infrared (IR). Pluto's haze possesses a bluish color, which suggests Rayleigh-type scattering by particles with radii smaller than visible wavelengths. However, the haze also has strong forward scattering, which is an indication of large particles[15]. Given these properties, the haze particles were thought to be fractal aggregates-highly porous and randomly shaped ~μm particles consisting of small ~10 nm spherical monomers, akin to those in Titan's atmosphere[10]. However, the non-negligible backscattering of Pluto's haze in the observations is inconsistent with a haze composed entirely of fractal aggregates[11]. The backscattering characteristics resemble those of Triton's haze, which is thought to arise from the combination of surface reflection and low-altitude clouds[9].

Separately constrained using observations from different instruments, several size distributions (log-normal, bimodal, power-law) together with surface reflection have been proposed to reproduce Pluto's haze observations[11,12,30], but degeneracy is significant. The backscattering has especially not been well investigated. Here, we conduct a joint retrieval using a combination of full phase observations covering a wide wavelength range obtained by all instruments onboard New Horizons that observed Pluto's haze and show that the backscattering of Pluto's haze originates from a smaller-size population of more compact haze particles. We further conclude that a unimodal distribution of haze cannot reproduce the observations. At least two populations with different fractal dimensions are necessary. A bimodal distribution is the simplest feasible solution, which is also the only one among several proposed scenarios that can currently be constrained.

Four New Horizons instruments measured the optical properties of Pluto's haze, including (1) UV extinction obtained by the Alice UV spectrograph[31] through solar occultation; (2) scattered light at a single visible wavelength at high and low phase angles captured by a wide-band camera, the Long Range Reconnaissance Imager (LORRI)[32]; (3) forward scattered IR spectra measured by a spectral imager, the Linear Etalon Imaging Spectral Array (LEISA)[33]; and (4) scattered light at four narrow visible and near-IR wavelength bands at high and low phase angles by a narrow-band camera, the Multispectral Visible Imaging Camera through its color filters (MVIC)[33]. Observations obtained by the MVIC panchromatic filters are not included due to data coverage and calibration issues (see Methods). Among these currently available observations of Pluto's haze, we select the ones with the highest quality and good resolution (>5 km), which are summarized in Table 1 and Fig. 1.

**Table 1 Summary of Pluto's haze observations.**

| Instrument | Wavelength (μm) | Altitude Range (km) | Phase Angle (degree) | Reference |
|---|---|---|---|---|
| Alice | 0.185 | 0–300 | Extinction | Young et al.[17] |
| LORRI | 0.608 | 0–100 | 19.5 | Cheng et al.[11] |
| | | 0–50 | 67.3 | |
| | | 0–75 | 148.3 | |
| | | 0–200 | 169.0 | |
| LEISA | 1.235–2.435 | 0–299 | 169.0 | Grundy et al.[29] |
| MVIC | 0.624 | 0–50 | 18.2 | This work |
| | 0.492 | | 38.8 | |
| | 0.861 | | 169.4 | |
| | 0.883 | | | |

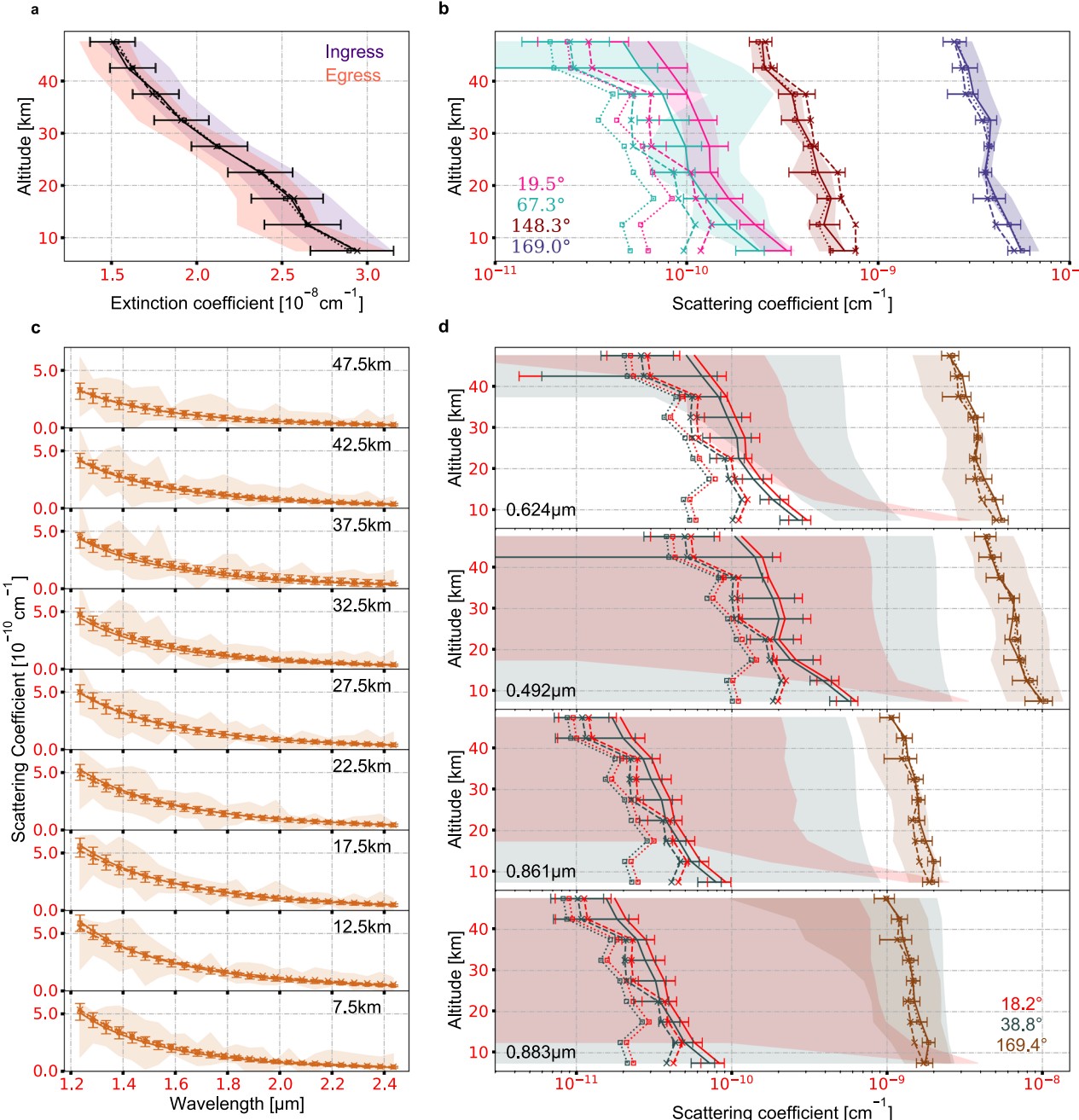

**Fig. 1 Comparison of different model haze scenarios to observations of Pluto's haze.** Haze observations obtained by instruments onboard New Horizons (shaded areas) are compared to best-fit model results for monodispersed fractal aggregates constrained using all observations (dashed lines with crosses), monodispersed fractal aggregates constrained using all observations except for the backscattering LORRI and MVIC data (dotted lines with squares), and a bimodal distribution of haze particles constrained using all observations (solid lines). Error bars show the 1-σ uncertainties of the bimodal distribution scenario. **a** UV extinction coefficient at 0.185 μm measured by the Alice spectrograph during solar occultation ingress (indigo) and egress (light red), taken from Young et al.[17]. **b** Local scattering intensity at 0.608 μm derived from LORRI images at four phase angles of 19.5° (pink), 67.3° (light green), 148.3° (dark red), and 169.0° (dark blue), processed using data from Cheng et al.[11] **c** Local scattering intensity spectrum as a function of altitude derived from LEISA observations at a phase angle of 169.0°, processed using data from Grundy et al.[29]. **d** Local scattering intensity at the 0.624, 0.492, 0.861, 0.883 μm (from top to bottom) wavelength bands derived using MVIC images at three phase angles of 18.2° (red), 38.8° (grey), and 169.4° (brown).

We focus on the lower 50 km of Pluto's atmosphere above the surface due to data coverage (Table 1), which includes its thin troposphere and an overlying thermal inversion. Haze in this region was observed by all aforementioned instruments, and therefore its optical properties are the best constrained. The morphology of haze particles in this region is representative of the final stage in the shaping of haze particles through microphysical processes in Pluto's atmosphere, just before they sediment onto the surface. The pressure level (0.5–1 Pa) of this region is comparable to 400–600 km altitude in Titan's atmosphere where the detached haze layer is imaged[34], and to altitudes of <50 km in Triton's atmosphere[35]. Despite higher temperatures on Titan and lower $CH_4$ abundances on Triton, the major formation pathways of hazes are expected to be similar[13,14].

The observed line-of-sight (LOS) integrated quantities (optical depth and I/F) are first converted to local optical properties

assuming spherical symmetry using the Abel transform for noisy data (see Methods). We then apply light scattering models and our retrieving algorithm to these local quantities (Methods). We consider the collective scattering effects of fractal aggregates, spherical haze particles, and surface reflection in ten scenarios (Supplementary Table 2, Methods), which are chosen based on a balance between the number of observations and the degree of freedom. The values of the parameters under each scenario and their uncertainties are obtained using the Markov-chain Monte Carlo (MCMC) approach[36,37] (Methods). We parameterize the morphology of fractal aggregates using three quantities, the fractal dimension ($D_f$), monomer radius ($r_m$) and number of monomers in each aggregate ($N_m$), which follow the relationship $N_m = (R_a/r_m)^{D_f}$ with aggregate effective radius ($R_a$). The fractal dimension describes the porosity of the aggregate and relates the change in size of the aggregate with its change in mass. Aggregates with larger $D_f$ are more compact. In the nominal forward model, we assume the haze particles are photochemically produced solids and use the complex refractive indices of "tholins", the laboratory analogues of Titan's haze[38], for Pluto's haze, motivated by similar atmospheric compositions between Pluto and Titan. The higher fraction of CO and organic ice condensation in Pluto's atmosphere may influence these optical properties[13,23,39], but our sensitivity study shows that differences in refractive indices would not significantly change the observables, and therefore would not affect our interpretation (Methods). We compute the optical properties of aggregates by adapting a light scattering model that was used for Titan's haze and well validate[10,40], and we use Mie theory[41] to compute scattering from spherical particles and monomers (Methods).

**Haze morphology.** As the simplest assumption, a monodispersed population of ~1 μm fractal aggregates consisting of ~20 nm monomers can reproduce the UV extinction, visible forward scattering, and the slope of the IR forward scattering spectra (Fig. 1). The aggregate effective radius is two times larger than those estimated in previous works[11]. However, as backscattering is orders of magnitude less intense than forward scattering for large fractal aggregates (Figs. 1 and 2), the monodispersed aggregates scenario underestimates the observed backscattering by a factor of ~3 (Fig. 1b), which is equivalent to a LOS I/F difference of ~$5 \times 10^{-3}$ in the visible scattering configuration in the lower 50 km. Moreover, tests of other possible unimodal scenarios (log-normal, power-law or exponential distribution of two-dimensional aggregates or spheres) suggest that unimodal distributions of either aggregates or spheres cannot reproduce the scattering intensities at the observed phase angles in the visible and the forward scattering spectrum in the IR with the given UV extinction (Methods, Supplementary Figs. 7 and 8). This is an indication of additional scattering sources.

Surface reflection was proposed as another scattering source[11], as it is suggested to be an important contributor in the backscattering configuration[21], and a combination of surface reflection and low-altitude clouds successfully explains the large backscattering observed for Triton's haze[9]. However, secondary scattering, sunlight first reflected off of the surface and then scattered once by the haze particles into New Horizons' instruments, is not sufficient in the case of Pluto. We conduct a quantitative estimation by adapting the Hapke model[42] to simulate Pluto's surface reflection (Methods), in line with the analysis of scattering properties from integrated Pluto images[21].

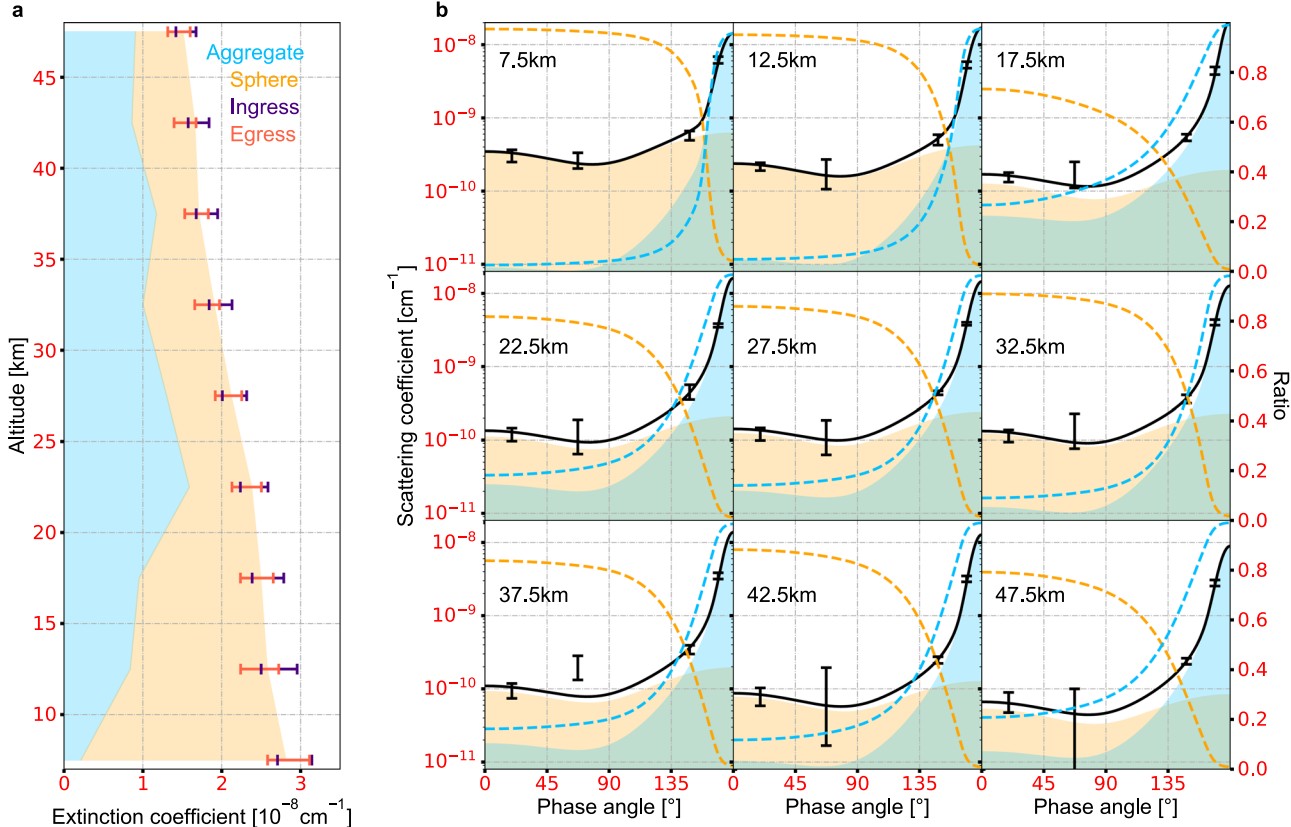

**Fig. 2 Contributions of fractal aggregates and spheres to UV extinction and scattering intensity in the visible. a** Observations obtained by the Alice spectrograph during ingress (indigo) and egress (red) of solar occultation are denoted as error bars and compared to the contributions from aggregates (blue) and spheres (orange). **b** Observations obtained by LORRI are denoted as black error bars and compared to the contributions from aggregates (blue shaded areas) and spheres (orange shaded areas). The solid black lines are the total contributions of aggregates and spheres, which are also the sum of the shaded areas. The colored dashed lines represent the ratio of contribution from each component.

An upper limit of $<2.5 \times 10^{-3}$ is derived for the I/F contribution from secondary scattering at the observation wavelength and phase angles when assuming that the incident and emission vectors are in the same specular plane (Supplementary Fig. 6), which is not sufficient to bridge the gap between the observed backscattering and that predicted by the monodispersed aggregates scenario.

Another possible scattering source is an additional population of small particles scattering in the Rayleigh regime. Forward and backward scatterings of these particles are comparable in strength, so that the total backscattering of the haze could be considerable when small particles are mixed with large aggregates (Fig. 2). Results of the joint retrieval under this scenario show that a combination of large two-dimensional aggregates and small spheres can reproduce all the observations (Fig. 1). The two types of haze particles have comparable UV extinctions while the aggregates dominate the forward scattering and spheres dominate the backscattering at visible wavelengths (Fig. 2). Vertical profiles of the retrieved parameters of the bimodal distribution are almost constant between 50 and 15 km (Fig. 3), indicating that the haze particle morphology does not have noticeable changes in this region. The larger-size population consists of ~1 μm two-dimensional aggregates with ~20 nm monomers (Fig. 3b, e), while the smaller-size population consists of ~80 nm spheres (Fig. 3b). The number density of the aggregates is ~0.3 cm$^{-3}$,

within an order of magnitude of previous estimations[11], while that of the spheres is around ~10 cm$^{-3}$ (Fig. 3d). Although the number densities differ by two orders of magnitude, the total masses of these two populations are almost the same (Fig. 3f). The mass of the aggregates is slightly greater, but within a factor of 2.

The consideration of all observations from UV to IR resolves the degeneracy in previous interpretations[11,12,30], allowing for more precise treatments of processes that are highly dependent on haze morphology and size, e.g., gaseous condensation and radiative transfer. Analysis of the visible phase functions and infrared forward scattering spectra with observed UV extinctions confirms the existence of at least two populations of haze particles with different fractal dimensions (Methods). Two monodispersed populations are the simplest feasible solution, though narrow sub-distributions centered at the two retrieved particle radii are possible.

**Formation pathway**. This result provides strong observational evidence of multi-mode haze formation processes in chemically reducing atmospheres. Given the similarities in composition and pressure between Pluto's lower atmosphere and the upper atmosphere of Saturn's moon Titan above 400 km, photochemical and microphysical modeling for producing the detached hazes on Titan can provide insights to the formation pathways of the

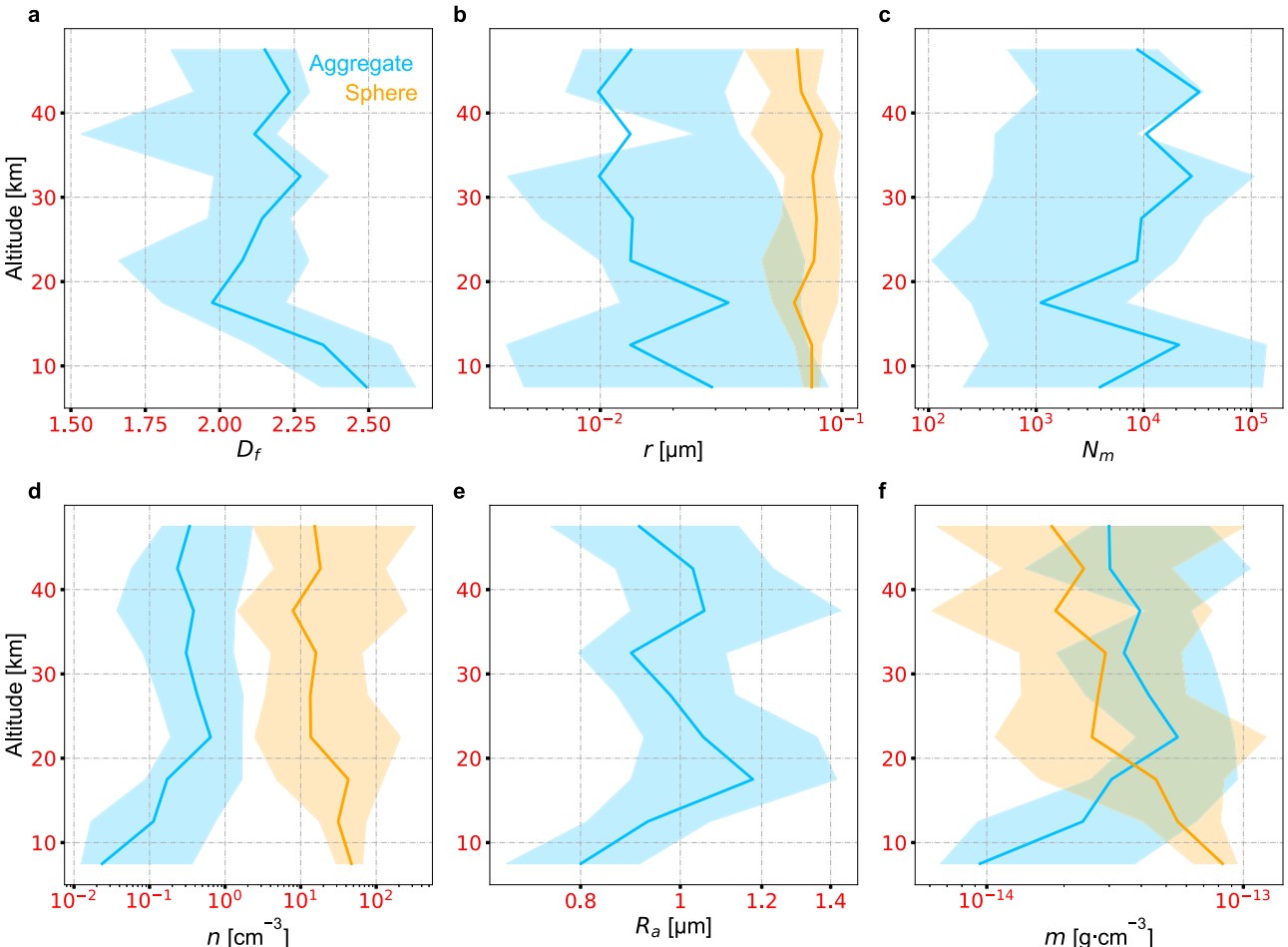

**Fig. 3 Retrieved profiles of haze parameters.** Under the scenario of the bimodal distribution, we retrieve vertical profiles for the (**a**) aggregate fractal dimension, (**b**) monomer/sphere radius, (**c**) number of monomers in each aggregate, and (**d**) haze particle number density, with which we derive profiles of the (**e**) aggregate effective radius, and (**f**) mass density assuming a material density of 1 g·cm$^{-3}$. The best-fit profiles are shown in the solid curves, while the 1-σ uncertainties of their posterior distribution functions are shown as shaded areas, with blue representing the aggregates population and orange representing the spheres population.

bimodal distribution of Pluto's haze. Numerical simulations[14] of Titan's upper haze show a fractal dimension transition from three (spheres) to two (fractal aggregates) near the pressure level ~0.5 Pa. The modeled Titan's haze at the dimensional transition region follows a bimodal distribution, including ~1 μm two-dimensional aggregates and small spheres with radii on the order of tens of nanometers. Our derived sizes and dimensions of the two populations of Pluto's haze agree surprisingly well with their simulated Titan's haze at a similar pressure level, indicating a similar formation mechanism. Moreover, the pressure level of the transition region in the model of Titan's haze was artificially selected for the purpose of producing two-dimensional aggregates in the lower atmosphere (<200 km) to match the observations. However, the exact physical condition that triggers the dimensional transition was unknown. Our Pluto data provides independent evidence from a similar world that particle dimensional transition indeed occurs in reducing atmospheres.

In the numerical simulations of Titan's haze[14], the spherical particles form and sediment from higher altitudes, where all particles follow a unimodal distribution. When these particles reach the transition pressure level, they begin coagulating into two-dimensional fractal aggregates. The cross sections of the aggregates are much larger than equivalent-mass spheres, so the number density of the larger-size population increases rapidly through coagulation once the dimensional change initiates, which results in a loss of medium size particles. The aggregates stop growing at ~1 μm due to Coulomb repulsion. In the case of Titan, the smaller-size population continues coagulating with large ~1 μm aggregates below the dimension transition region, ultimately being subsumed into the large fractal aggregates population. In contrast, the atmosphere of Pluto is much thinner with a surface pressure of ~1 Pa. As such, the haze particles reach Pluto's surface in the middle of dimensional transition and the bimodal distribution is maintained. Moreover, the sharp decrease of the atmospheric temperature[18] from ~100 to 40 K in the lower 15 km could slow down or freeze the process of coagulation and may introduce another dimensional change with rapid condensation of gaseous species[23] and possibly the collapse of porous fractal aggregates, which is suggested by the significant increase of fractal dimension of the larger-size aggregate population below 15 km altitude (Fig. 3a).

**Implications**. Our results can also shed light on the formation of Titan's detached haze layer. Other than the photochemical origin scenario[14], a dynamical origin[43] has also been proposed. It presumes a large-scale circulation that lifts the haze particles from higher-pressure regions to explain the existence of Titan's detached haze. As Pluto's haze is near its surface, considerable mass transport from a reservoir below is unlikely and thus the observed multi-mode on Pluto favors the photochemical origin. For the case of Titan, we suggest that constraining the bi-modality of Titan's detached haze is critical for distinguishing between the photochemical and dynamical origin scenarios. Images taken with the Imaging Science Subsystem (ISS) onboard the Cassini spacecraft would provide fruitful observations over a number of viewing configurations and different seasons[34]. Backscattering observations of the detached haze layer would contain information about a second population of small spheres.

Reanalysis of observations of Triton's haze is also crucial in light of our results. Two types of spherical particles, hazes and clouds, were proposed at different latitudes to explain the observed scattering obtained by the Voyager 2 wide-angle camera, together with surface reflection[9]. Haze particles are assumed to be spheres, but fractal aggregates are more likely given the formation mechanism[13,23]. Moreover, the observed large

backscattering may be due to a second population of small particles, which could contribute orders of magnitude more to the observed I/F than the secondary scattering of surface reflected light.

The discovery of the small-sphere haze particle population has significance in improving our understanding of the radiative processes in Pluto's atmosphere, and therefore Pluto's energy budget. Pluto's unexpectedly low atmospheric temperature was well explained by efficient radiation by ~1 μm haze particles and effective collision between these particles and gas molecules[28]. Solar energy absorbed by gases is transferred to haze particles rapidly through collision and radiated away from Pluto, which is proposed as the dominant energy pathway in Pluto's atmosphere. However, compared to the ~1 μm aggregates, smaller particles with radii on the order of tens of nanometers have shorter radiative relaxation timescales but much longer collisional heat-transfer timescales with gases. Compared to ~1 μm aggregates, the ratio of these two timescales for ~80 nm particles differ by 2–3 orders of magnitude, which leads to the radiation timescale being smaller than the collision timescale. In other words, the radiative cooling of smaller particles can be so efficient that the heat-transfer between the particles and gases through collision is not sufficient to keep their temperatures the same. This could result in the small particles being cooler than the ambient atmosphere. The larger-size particles are still the key medium transferring heat between gases and particles, but the dominant factor of radiative cooling needs reevaluation. The more efficient radiative cooling of smaller particles may result in another peak and/or a steeper slope in the mid-infrared emission spectrum of Pluto, different from previously predicted[28], which can be further investigated by future missions, e.g., through the Mid-Infrared Instrument on the James Webb Space Telescope[44].

## Discussion

Condensation of gaseous species onto haze particles may influence their evolution in the atmosphere and their interactions with visible and infrared radiation. In the lowest 50 km of Pluto's atmosphere, where the pressure varies between 0.1–1 Pa, haze particles are expected to contain an organic ice component, similar to the predictions for Triton's atmosphere[23]. In contrast, Titan's atmospheric temperature is higher than 150 K at the same pressure level, which would inhibit gas condensation. Therefore, the hazes in Pluto's and Triton's atmosphere may have larger scattering and less absorption than "tholins". However, such gas condensation processes could not simultaneously reproduce the large forward and backward scattering with the given UV extinction observed at Pluto. With the inclusion of gas condensation, numerical simulations[23] show that fitting the UV extinction results in an underestimation of both the forward and backward scattering I/Fs at visible wavelengths by a factor of 2–3. Another scattering source is still necessary. Moreover, analyses using disk-integrated images of these three celestial bodies[21,45,46] suggest that even with condensation, the phase function of Pluto's haze is comparable with that of Titan, but the haze of Triton is bluer likely due to intense condensation of neutral ice, such as $N_2$. To test the influence of organic ice, we conducted a sensitivity study of two-dimensional aggregates with different optical properties (Methods). This represents an upper limit for the extent to which ice condensation or other organic composition can influence the observables. The result shows that the difference in scattering intensity at the wavelengths and phase angles considered here due to the different refractive indices is negligible. The observed quantities, especially the phase functions of fractal aggregates, are not sensitive to the difference in optical constants between organic ice and "tholins". A second component of small

spheres is still required to explain the observed backscattering of Pluto's haze. Thus, the influence of gas condensation is not large enough to alter our inferred pathway of haze formation.

As Pluto's atmosphere is in sublimation/deposition equilibrium with surface $N_2$ ice[15,17], its surface pressure is expected to vary significantly over seasons[47,48]. Small changes of the atmospheric temperature (e.g., a few K), driven by the eccentric orbit (e ~0.25), can result in orders of magnitude differences in surface pressure. Numerical simulations[48] show that Pluto could have a minimum surface pressure between $10^{-3}$–0.3 Pa on its current orbit near aphelion, which is large enough to maintain $CH_4$ photochemistry and therefore haze formation[17]. However, the dimensional transition region of Pluto's haze (0.5–1 Pa) will move downward and finally disappear when Pluto moves farther from the Sun, which may result in a significant difference in the size of haze particles depositing onto Pluto's surface over seasons. Also, when Pluto moves away from perihelion, the $CO/CH_4$ ratio is expected to increase, and therefore so does the fraction of oxygen atoms in the photochemically produced haze particles, as more $CH_4$ condenses out of the atmosphere onto the surface than CO when the temperature decreases. Thus, while understanding the morphology of haze particles and its influence on Pluto's system is important, the current interpretation is only representative of New Horizons' flyby in 2015. As Pluto's atmosphere evolves, so will the haze distribution and composition, allowing future observations and missions to capture different stages of organic haze evolution near Pluto's surface.

## Methods

**MVIC data processing.** Details of the MVIC instrument design and operation are given in Reuter et al.[33]. Science operation of MVIC is under time delay integration (TDI) mode with two panchromatic and four color arrays. We select the observations obtained by the color arrays as they contain Pluto haze's spectral characteristics. Level 2 data is used for analysis in this work, whose identifier on NASA PDS is New Horizons MVIC Pluto Encounter Calibrated Data v3.0[49]. The data contains bias-subtracted, flattened images, but it does not include corrections for scattered light, cosmic rays, and geometric and motion distortion. We follow the New Horizons SOC to Instrument Pipeline ICD to convert the calibrated data number (DN) to I/F, then use the SPICE system[50] to compute the geometry with navigation data (Supplementary Fig. 1). With the geometry information, we determine the resolution and mean phase angle of each image. Three observations with resolution better than 5 km/pixel are selected (Supplementary Table 1). As the geometry inferred by the navigation data does not perfectly locate Pluto in each image (Supplementary Fig. 1), we conduct a further correction, as the haze altitude needs to be accurate. A one-pixel offset may result in an altitude difference as large as 5 km, comparable to the bin interval selected in this work. To correct the Pluto location in each image we use established techniques for determining the geometric limb of Pluto and Charon. For frontlit images we use Method A from Nimmo et al.[51]. For backlit images we select the points with the largest brightness gradient around the sun-lit edge (Supplementary Fig. 1), then fit a circle to these points, which serves as Pluto's edge. Other geometries are offset according to the correction. Because MVIC is a scanning camera, there are distortions in Pluto's long-wavelength shape from a perfect sphere. To minimize the impact of these effects we select regions where the edge of Pluto is best defined for our analysis (Supplementary Fig. 1).

Stray light is not negligible in the images, as in LORRI observations[11], and it is especially important in the backscattering configuration when the disk brightness is much higher than the haze. Since stray light is an instrument effect, it is a function of pixel distance above the limb of celestial bodies, which can also be seen around the airless Charon (Supplementary Fig. 2). I/F profiles of Pluto's haze in LORRI observations are corrected by subtracting the normalized stray light above Charon's limb[11]. Here, we conduct the same correction: we compute the moving averages of I/F profiles as a function of pixel distance from Charon's limb, then subtract them from Pluto's. As Charon appears in only one of the observations (MET 0299162512), and the instrument effect should stay the same during the fast flyby, we assume the stray light influence is the same among images obtained by the same color array at different times, and apply the correction individually for each array (Supplementary Fig. 2). The corrected I/F profiles are used in the haze property retrievals.

Six images, five for Pluto and one for Charon, were taken through the two MVIC panchromatic arrays with resolution better than 5 km. Charon was only imaged once by the first panchromatic array (Pan 1, MET 0299180334), which results in the failure of calibrating the three observations by the second array (Pan 2). Also, in this image Charon is at a phase angle of ~85.2°, and its edge is not as

sharp as that imaged through the color arrays in backscattering configurations, which introduces considerable uncertainty, and also cross validation is lacking as only one observation is obtained. Since the panchromatic array has a similar wavelength-dependent response as that of LORRI[32,33], and the four MVIC color arrays also cover this wavelength range, we do not include the two images of Pluto made by Pan 1. Moreover, one of the two images only contains part of Pluto, which results in the difficulty determining Pluto's center.

**Vertical profile conversion.** In remote sensing of planetary atmospheres, observables are usually line-of-sight (LOS) integrated quantities, necessitating the conversion of these observables to local quantities for analyses of physical and chemical properties of the atmosphere. We use the Abel inverse transform of noisy data to obtain vertical profiles of the local quantities.

In the solar occultation by Alice, the observable is LOS optical depth.

$$\tau_{LOS}(r) = \int_{-\infty}^{+\infty} n(s)\sigma_{ext}(s)\,ds \tag{1}$$

where $\tau_{LOS}$ is the LOS optical depth; $r$ is the distance of Pluto's center to LOS; $n$ is the local number density; $\sigma_{ext}$ is the extinction cross section; and $s$ is the path along LOS. In the limb scattering geometry of the other three instruments, the observed I/F is an integration of local scattering intensity.

$$I/F(r) = \int_{-\infty}^{+\infty} \frac{1}{4} P(\cos\theta) n(s)\sigma_{sca}(s)\,ds \tag{2}$$

where $\sigma_{sca}$ is the scattering cross section; $\theta$ is the scattering angle; $P$ is the phase function, which is normalized to $4\pi$. Given the same form of Eqs. (1) and (2), they can be unified as follows.

$$N(r) = \int_{-\infty}^{+\infty} D(s)\,ds \tag{3}$$

where $N$ is the observable and $D$ is the corresponding local quantity. Assuming spherical symmetry, $D$ is a function of $r$, and the equation becomes an Abel integral.

$$N(r) = 2\int_{r}^{+\infty} D(r') \frac{r'}{\sqrt{r'^2 - r^2}}\,dr' \tag{4}$$

where $r'$ is the distance of Pluto's center to the point of each $D$ in the integral. This equation shows that $N$ only depends on $D$ at altitudes higher than the impact parameter. The Abel inverse transform suggests an exact solution[52].

$$D(r') = -\frac{1}{\pi}\int_{r'}^{+\infty} \frac{dN(r)/dr}{\sqrt{r^2 - r'^2}}\,dr \tag{5}$$

However, the exact solution is not a good option in the inversion of noisy data, as the derivative of $N$ in the integral is sensitive to noise. Therefore, instead of using the exact solution, we rewrite Eq. (4) with discrete altitude bins and solve the problem through linear regression.

$$N_i = 2\sum_{j=i}^{m-1} D_j \int_{r_j}^{r_{j+1}} \frac{r'}{\sqrt{r'^2 - r_i^2}}\,dr' \tag{6}$$

where $i$ and $j$ are the indices of altitude bins; $m$ is the total number of bins; $r_j$ and $r_{j+1}$ are the lower and upper boundary of the $j$-th altitude bin; and $N_i$ and $D_j$ are the corresponding LOS and local quantities at the $i$-th and the $j$-th altitude bins, respectively. In this case, all the integrals form a matrix **A**, which consists of element $A_{ij}$ for each pair of $(i, j)$ with $i \le j$.

$$A_{ij} = \int_{r_j}^{r_{j+1}} \frac{r'}{\sqrt{r'^2 - r_i^2}}\,dr' \tag{7}$$

**A** is an upper triangular matrix as $A_{ij} = 0$ when $i > j$. Including the noise in the data, there is a linear relationship between vectors of $N$ and $D$.

$$\mathbf{N} = \mathbf{A}\bullet\mathbf{D} + \boldsymbol{\varepsilon} \tag{8}$$

where **N** and **D** are column vectors, which have $N_i$ and $D_j$ as the elements; and $\boldsymbol{\varepsilon}$ is the vector with noise for each corresponding $N_i$. Then the linear problem can be solved as follows.

$$\mathbf{K} = \left(\mathbf{A}^T\mathbf{C_N}\mathbf{A}\right)^{-1}\bullet\left(\mathbf{A}^T\mathbf{C_N}^{-1}\right) \tag{9}$$

$$\mathbf{D} = \mathbf{K}\bullet\mathbf{N} \tag{10}$$

$$\mathbf{C_D} = \mathbf{K}\bullet\mathbf{C_N}\bullet\mathbf{K}^T \tag{11}$$

where $\mathbf{C_D}$ is the covariance matrix of **D**; $\mathbf{C_N}$ is the covariance matrix of **N**. The square root of the diagonal elements in $\mathbf{C_D}$ serves as the uncertainties of local quantities. Inversion through linear regression does not contain derivatives, so it is more robust against noise. Further regularization[53], which is not included in this work as the result is already acceptable, can also be added to decrease the influence of noise.

The transform requires the upper boundary of the altitude bins to be sufficiently high so that the truncation from infinity to $r_m$ in Eq. (6) can be neglected. However, some of the observations have limited altitude ranges due to the instrument field of

view. The I/F profile obtained by LORRI at a phase angle of 67.3° is limited to the lower 50 km, which is the smallest altitude range among all observations, and therefore constrains the altitude range where we can conduct our analysis. As Pluto's haze extends up to ~200 km above the surface at visible wavelengths with a density scale height of ~50 km near surface[15], haze above the available observation range is not negligible. Due to the small radius of Pluto (1190 km)[18], the change of scale height with gravity needs to be considered in the extrapolation. We assume an exponential decay in geopotential of the local density, as in Young et al.[17]. The first-order approximation is as follows[54].

$$H = H_0 \frac{r^2}{r_0^2} \tag{12}$$

$$D = D_0 e^{-\frac{r_0}{H_0}\left(1 - \frac{r_0}{r}\right)} \tag{13}$$

$$N = N_0 e^{-\frac{r_0}{H_0}\left(1 - \frac{r_0}{r}\right)} \left(\frac{r}{r_0}\right)^{\frac{3}{2}} \frac{1 + \frac{9}{8}\frac{H}{r}}{1 + \frac{9}{8}\frac{H_0}{r_0}} \tag{14}$$

where $H$ is the scale height; the subscript 0 indicates the surface value of corresponding variables. Through this approach, we extrapolate $N$ to 2000 km above Pluto's surface by fitting Eq. (14) to observables. The ranges selected for the fitting are between 25 km to the highest valid altitude for LORRI, 25–75 km for LEISA, and 15–50 km for MVIC observations. The conversion of Alice observations is done and published in Young et al.[17], whose results are adapted in this work.

The extrapolation leads to two distinct altitude regions in our analysis, which requires Eq. (8) to be rewritten,

$$\begin{bmatrix} \mathbf{N}_1 \\ \mathbf{N}_2 \end{bmatrix} = \begin{bmatrix} \mathbf{A}_{11} & \mathbf{A}_{12} \\ 0 & \mathbf{A}_{22} \end{bmatrix} \bullet \begin{bmatrix} \mathbf{D}_1 \\ \mathbf{D}_2 \end{bmatrix} + \begin{bmatrix} \boldsymbol{\varepsilon} \\ 0 \end{bmatrix} \tag{15}$$

where subscript 1 denotes the corresponding variables at altitudes with valid observations, while subscript 2 denotes those of the extrapolated altitudes. We assume that the extrapolation has no noise. Combining Eq. (15) with (9–11), we can finally derive the required local quantity and its uncertainty.

$$\mathbf{D}_2 = \left(\mathbf{A}_{22}^T \mathbf{A}_{22}\right)^{-1} \bullet \left(\mathbf{A}_{22}^T \mathbf{N}_2\right) \tag{16}$$

$$\mathbf{K}_1 = \left(\mathbf{A}_{11}^T \mathbf{C}_{\mathbf{N}_1} \mathbf{A}_{11}\right)^{-1} \bullet \left(\mathbf{A}_{11}^T \mathbf{C}_{\mathbf{N}_1}^{-1}\right) \tag{17}$$

$$\mathbf{D}_1 = \mathbf{K}_1 \bullet (\mathbf{N}_1 - \mathbf{A}_{12}\mathbf{D}_2) \tag{18}$$

$$\mathbf{C}_{\mathbf{D}_1} = \mathbf{K}_1 \bullet \mathbf{C}_{\mathbf{N}_1} \bullet \mathbf{K}_1^T \tag{19}$$

where $\mathbf{C}_{\mathbf{D}1}$ and $\mathbf{C}_{\mathbf{N}1}$ are the covariance matrices of $\mathbf{D}_1$ and $\mathbf{N}_1$, respectively. $\mathbf{D}_1$ corresponds to the UV extinction coefficient($n\sigma_{ext}$), or local scattering intensity ($\frac{1}{4}Pn\sigma_{sca}$) in the integral of Eqs. (1) and (2), respectively.

**Scattering models**. Light scattering by spherical haze particles and monomers in fractal aggregates is computed using Mie theory[41], as their radii are comparable to some of the observation wavelengths. For the fractal aggregate particles, we use the scattering model described by Tomasko et al.[10] to estimate their phase functions and cross sections. The model was originally developed to constrain haze particle properties in Titan's atmosphere. Given the similarity in atmospheric composition, the haze particles are likely to be similar. The scattering model uses empirical phase functions derived from averaging exact results of randomly produced aggregates, which significantly reduces the computational time, resulting in the feasibility of retrieval. It computes the phase functions and cross sections at a given wavelength with three parameters describing an aggregate ($D_f$, $r_m$, and $N_m$) and the complex refractive index. An illustration of how $D_f$ influences the aggregate morphology is given in Supplementary Fig. 3. Another scattering model[55] was tested but not used in this work due to its omission of polarization, which is not negligible when monomer number becomes large (~1000), although its disagreement with Tomasko et al.[10] is less than 20% near our retrieval results. The scattering model we used in this work is rigorously tested at $D_f = 2$ and mostly at $N_m < 10^3$, but testing has shown that perturbation of $D_f$ is allowed (1.5 < $D_f$ < 2.5), and the extrapolation of $N_m$ to ~$10^4$ is reasonable given the linear relationship in log-log scale between cross section and monomer number at the larger end of aggregate sizes[10].

**Retrieving algorithm**. We use the Markov-chain Monte Carlo (MCMC) method[36] as the parameter searching tool. It derives the posterior probability density function (PDF) of each parameter by comparing the posterior probabilities of proposed parameter sets. The cost function with posterior probability ($p$) is defined as

$$\ln(p) = -\frac{1}{2}\sum_i \frac{(X_i - \mu_i)^2}{\sigma_i^2} \tag{20}$$

where $\mu_i$ and $\sigma_i$ are the value and uncertainty of the $i$-th observation, respectively, and $X_i$ is the modeled $i$-th observation computed using a given proposed parameter set during one MCMC attempt. The Python package emcee[37] is used to implement

the parameter searching algorithm. We initiate the MCMC process with 40 chains and flat priors for all parameters, and run the parameter search for 1000 steps. The last 500 steps are selected for the result analysis, which are considered to be in the equilibrium state, as most of the chains converged after only 200 steps.

An advantage of using MCMC as the parameter searching tool is that it gives the extent to which a parameter can be constrained. The algorithm does not require any assumptions of the shape of posterior PDFs, which are necessary for computing gradients in other approaches (e.g., the Levenberg–Marquardt algorithm[56]). This is critical when the observation is barely sufficient for constraining parameters as shown in the Pluto haze observations, where well-constrained parameters have Gaussian-like PDFs, and poorly constrained ones have PDFs varying over large ranges (Supplementary Fig. 4). MCMCs can also avoid converging to local minima, which is important in this work as the relationship among the four fractal aggregate parameters are not linear, and therefore multiple local minima are expected.

**Surface reflection model**. Surface reflection may have non-negligible contributions to the observed haze brightness, as suggested by studies of Pluto's and Triton's hazes[21,45,46]. We estimate this contribution in one of the tested scenarios. In line with Hiller et al.[21,45,46] we use the Hapke model[42] to simulate Pluto's surface reflection. Due to the small optical opacity of Pluto's atmosphere, attenuation of incoming and reflected light by the atmosphere are neglected. We assume Pluto's surface is a perfect sphere composed of uniform isotropic scatters, so that the reflected light follows

$$I(\mu_0, \mu) = J\frac{w}{4\pi}\frac{\mu}{\mu_0 + \mu}H(\mu_0)H(\mu) \tag{21}$$

where $I$ and $J$ are the reflected and incoming intensities, respectively; w is the single scattering albedo of surface materials; $\mu_0$ and $\mu$ are the cosines of solar and viewing zenith angles, respectively; and $H(\mu)$ is the Hapke function

$$H(\mu) = \frac{1 + 2\mu}{1 + 2\sqrt{1 - w\mu}} \tag{22}$$

We use the Hierarchical Equal Area isoLatitude Pixelization method (HEALPix)[57] to discretize Pluto's surface for the reflection computation. This technique divides the surface into pixels with the same areas distributed uniformly on the sphere. The parameter $N_{side}$ in HEALPix is set to 16, which results in a 3072-pixel map with a spatial resolution of ~4°. Unlike in the retrieval of the surface and haze properties of Pluto in Hillier et al.[21], we simplify the model to estimate the upper limit of the contribution from secondary scattering, sunlight first reflected off of the surface and then scattered once by the haze particles into New Horizons' instruments, by assuming that the incident and emission vectors are in the same specular plane.

**Test of scenarios**. Given the valuable but limited observations, degeneracy appears when the number of free parameters is too large. Therefore, we tested a number of haze morphology scenarios in the retrieval (Supplementary Table 2), which are compromises between fitting all the observations and limiting the degree of freedom. The scenarios included in this work are (1) monodispersed fractal aggregates with variable dimension, (2) monodispersed fractal aggregates with variable dimension and surface reflection, (3) monodispersed fractal aggregates with variable dimension and monodispersed spheres, (4) two populations of monodispersed spheres, and (5)–(10) log-normal, power-law or exponential distribution of two-dimensional aggregates or spheres. As shown below, scenario (3) of a bimodal distribution consisting of large fractal aggregates and small spherical particles is the only one that can fit all the observations at various scattering angles and wavelengths.

Under scenario (1), we assume that there is a monodispersed population of aggregate haze particles at each altitude and include four free parameters. Three of them ($D_f$, $r_m$, $N_m$) describe the morphology of haze particles, and the fourth one ($n_a$) is the aggregate local number density. Vertical profiles of these four quantities that best fit the observations are shown in Supplementary Fig. 5, and the corresponding simulated observables are given in Fig. 1, where "best-fit" is defined as when the posterior probability as defined in Eq. (20) is maximized. The retrieval shows that the assumption of monodispersed fractal aggregates cannot reasonably fit the observations, as it underestimates the backscattering (Fig. 1b) due to the forward-scattering dominated phase functions of fractal aggregates. This is consistent with the discrepancy suggested by Cheng et al.[11] that one population of fractal aggregates or spheres alone cannot explain both the forward and backward scattering at visible wavelengths and the UV extinction.

As fractal aggregates tend to underestimate the backscattering, and surface reflection usually has more intense backscattering than forward, under scenario (2), we test whether surface reflection can fill in the gap between the observed backscattering intensity and that scattered by aggregates. The observed intensity above Pluto's limb is assumed to consist of two parts: (1) light scattered once by haze particles (primary scattering) and (2) light reflected by Pluto's surface and then scattered by haze particles (secondary scattering).

As surface reflection is negligible when the viewing zenith angle is large, we first omit the backscattering observables and only use the forward ones (LORRI at 148.3° and 169.0°, LEISA, and MVIC at 169.4°) together with the UV extinction to constrain the haze morphology. The resulting profiles of the four free parameters

$(D_f, r_m, N_m, n_a)$ and comparison with observations are given in Supplementary Fig. 5 and Fig. 1, respectively. The simulated forward scattering of haze particles agrees with observations well and all four parameters are well-constrained across all the altitudes. The best-fit haze particles are ~1 μm two-dimensional aggregates with ~20 nm monomers at most of the altitudes. However, the backscattering by these particles is underestimated. The local scattering intensity of haze particles is less than the observed value by a factor of two (Fig. 1b), which results in a LOS I/F difference of ~$5 \times 10^{-3}$ in the lower 50 km. Therefore, we test the inclusion of surface reflection to try to fill in the gap.

With the fixed retrieved haze morphology, we estimate the upper limit of the secondary scattering by computing the I/F assuming the incident and emission vectors are in the same specular plane. For each discretized point along the LOS, the secondary scattering is computed by summing up the reflected light from the pixelated Pluto's surface multiplied by the haze scattering phase function with corresponding geometry. Integrating the secondary scattering along the LOS then provides the observables.

Our results (Supplementary Fig. 6) suggest that, even with the highest surface single scattering albedo ($w = 1$), the maximal value of the LOS-integrated secondary scattering, which is near ~90° phase angle, is smaller than half of the gap between the haze primary scattering and observed I/F. Moreover, the maxima at the observed phase angles of LORRI (19.5°, 67.3°, 148.3°, and 169.0°) are around or less than $10^{-3}$, so the upper limit of the secondary scattering is at least one order of magnitude less than the required backward scattering.

As monodispersed fractal aggregates and surface reflection cannot reproduce the observed large backscattering, we consider a bimodal distribution of two haze particle populations with different sizes under scenario (3). Besides the aggregates as described in scenario (1), we include a population of small spheres, which are parameterized with two variables, radius ($R_s$) and number density ($n_s$). Therefore, six free parameters in total are considered in the retrieval. The vertical profiles of these six parameters, as determined by the MCMC, are shown in Fig. 3, and their PDFs at one of the altitudes (22.5 km) are given in Supplementary Fig. 4. Similar to those in scenario (2), we obtain a population of ~1 μm two-dimensional aggregates with ~20 nm monomers across the entire considered altitude region, as well as a population of spheres with radii of ~80 nm. These two radii in the bimodal distribution are similar to those simulated in Titan's atmosphere[14]. Comparison between the modeled observables and observations (Fig. 1) indicates that, under this scenario both the large forward and backward scattering could be explained, along with the given UV extinction. The two types of haze particles have comparable UV extinctions, with each dominating one of the forward and backward scattering in the visible (Fig. 2). Some "apparent" disagreements are seen for the MVIC backscattering observations (Fig. 1b), which are due to their large uncertainties, as the bandpass of the MVIC color filters are much smaller than that of LORRI[32,33]. The derived backscattering intensities are still mostly within the 1-σ uncertainties.

Seven other particle size distributions were also tested: (4) bimodal distribution of spheres (particles with $D_f = 3$), (5) log-normal distribution of spheres, (6) power-law distribution of spheres, (7) exponential distribution of spheres, (8) log-normal distribution of two-dimensional aggregates (particles with $D_f = 2$), (9) power-law distribution of two-dimensional aggregates, and (10) exponential distribution of two-dimensional aggregates. We quantify the goodness of fit using the maximal probability that can be reached under each scenario as defined in Eq. (20). All of the seven scenarios presented here show goodness of fit far worse than the bimodal distribution of large aggregates and small particles (Supplementary Table 2, Supplementary Fig. 7).

Scenario (4) of a bimodal distribution of spheres contains four free parameters. They are the sphere radii and the two corresponding number densities. Scenario (5) contains three free parameters, which are the two parameters defining the log-normal distribution and the total number density. We include 30 size bins of spheres, covering a radius range from 1.3 nm to 1.0 μm. Each bin is assumed to have particles with twice the mass of the previous, so the ratio of the radii in two consecutive bins is $\sqrt[3]{2}$. Therefore, the number density of particles in each bin is defined as

$$n_i = n_0 \left( CDF(r_{2i}) - CDF(r_{1i}) \right) \quad (23)$$

where $i$ is the bin index; $r_{1i}, r_{2i}$ are the radii at the lower and upper boundary of the $i$-th bin, respectively; $n_i$ is the number density of particles in the $i$-th bin; $n_0$ is the total number density. CDF is the cumulative density function, which, for the log-normal distribution, is defined as

$$CDF_{LN}(r) = \frac{1}{2} + \frac{1}{2} erf \left( \frac{ln(r) - \mu}{\sqrt{2}\sigma} \right) \quad (24)$$

where $r$ is the particle radius; $\mu$ and $\sigma$ are the mean and standard deviation of the logarithm of radius, respectively; and erf is the error function. Under this scenario, $n_0$, $\mu$ and $\sigma$ are the three free parameters. Scenario (6) is the same as scenario (5) except for the CDF, which is defined as

$$CDF_{PL}(r) = 1 - r^{1-p} \quad (25)$$

where $p$ is the power describing how fast the number density decreases with size. As the CDF is defined using one parameter, this scenario has two free parameters

($n_0$ and $p$). Similarly, scenario (7) has a CDF defined as

$$CDF_{Exp}(r) = 1 - e^{-\alpha r} \quad (26)$$

where $\alpha$ is the exponent describing the decrease of number density with size. This scenario also has two free parameters ($n_0$ and $\alpha$).

Scenarios (8)–(10) are the same as scenarios (5)–(7), respectively, except for the particle bins. A monomer size ($r_m$) is assumed and fixed during each retrieval, and we tested a group of retrievals with monomer size from 1 nm to 0.1 μm. The monomer number ratio is 2 between consecutive bins, thereby maintaining the mass doubling with successive bins. As the fractal dimension is fixed to be 2, the effective radius ratio between consecutive bins is $\sqrt{2}$. The smallest size bin contains 2 monomers, and the largest contains $2^{30}$, which is equivalent to an effective radius ~$3 \times 10^5$ times that of the monomer. The optimal monomer sizes to reach the maximal probabilities are 20, 20, and 30 nm under the scenarios (8)–(10), respectively.

To quantify the goodness of fit of the proposed scenarios, we compare the maximum posterior probability that can be reached under each scenario. The combination of parameters that results in such maximal posterior is then defined as the best-fit. An example of the best-fit under scenario (3) is shown as the black lines in Supplementary Fig. 4. The best-fit value of each parameter may not be the same as that at the maximal probability of its individual posterior distribution. For the purpose of illustration, we show the goodness of each scenario by the negative of the natural logarithm of the posterior probability, $-\ln(p)$, whose average is shown in Supplementary Table 2 and altitude-dependent values are shown in Supplementary Fig. 7. The scenario with a bimodal distribution of aggregates and spheres has significantly smaller $-\ln(p)$, which corresponds to orders of magnitude larger probability, than all the others.

Several haze particle size distributions have been proposed and fit to some of the observations considered in this work. Gladstone et al.[15] suggested >0.1 μm aggregate particles consisting of ~10 nm monomers based on order-of-magnitude estimations using observations at a few visible wavelengths and phase angles obtained by LORRI and MVIC. Gao et al.[12] conducted a microphysical simulation of haze particle formation and proposed that a log-normal distribution centered at 0.1–0.2 μm can well explain the observed UV extinction measured by Alice. However, Cheng et al.[11] found that a single population may not be able to explain the combination of observations at visible wavelengths from LORRI and UV extinction from Alice. A log-normal distribution of spherical haze particles centered near 0.5 μm together with a surface albedo of 0.5 can fit the haze phase function in the visible, but the associated UV extinction is too small, while ~0.15 μm aggregates explain the UV extinction well but underestimate the backscattering. Kutsop et al.[30] conducted a more detailed retrieval and were able to simultaneously explain the Alice UV extinction data and MVIC observations at three of the color filters and at seven-phase angles across a much greater altitude range (0–500 km) but at lower altitude resolution than in our work. However, they were unable to differentiate between bimodal and power-law haze particle size distributions.

In addition to observations in the UV and visible, scattering spectra of Pluto's haze was also observed in the IR by LEISA, which was not included in the aforementioned works. Here we discuss how consideration of LEISA data addresses the degeneracy in haze size distributions. Supplementary Fig. 8 shows the scattering intensities of two-dimensional aggregates and three-dimensional spheres in the visible (LORRI) and IR (LEISA) with unit UV extinction (Alice), assuming tholin refractive indices. The monomer radius in the aggregates is assumed to be our retrieved value of 20 nm. The ratio of scattering intensity in the visible to the UV extinction for aggregates (Supplementary Fig. 8a) shows that large aggregates (>0.5 μm) can reasonably explain the observed forward scattering, but also that aggregates of all sizes underestimate the backscattering. This is consistent with Cheng et al.[11] and also our scenario (1) and indicates that no distribution of two-dimensional aggregates is sufficient to explain the observed backscattering. In our bimodal distribution, the sizes of the large aggregates are mainly constrained by forward scattering and especially the IR spectra, as forward scattering at longer wavelengths is overwhelmingly determined by the larger-size population (Supplementary Fig. 8c). Aggregates with ~1 μm radius are the best fit to the IR data, while the contribution from small particles is negligible.

The scattering phase functions of spheres are more symmetric than aggregates with the same radii, and less asymmetric than those observed (Supplementary Fig. 8b). Therefore, though the large forward scattering of Pluto's haze is unlikely to be due to spheres, they can provide sufficient backscattering in the visible for a given UV extinction due to their small cross sections (Supplementary Fig. 8b). As such, the particle sizes of the small-spheres population in our bimodal distribution are primarily constrained by the backscattering observations. Here, the IR data is able to rule out different spheres-only size distributions (i.e., scenarios (5)–(7)), as the slope of the ratio of IR scattering intensity to the UV extinction for spheres as a function of wavelength are all steeper than the observations (Supplementary Fig. 8d). Therefore, we are able to break the degeneracy of different size distributions found by Kutsop et al.[30]

In summary, due to the weak backscattering of two-dimensional aggregates and steeper IR forward scattering spectra of three-dimensional spheres, the observed optical properties of Pluto's haze cannot be explained by either one of these populations alone, even with non-monodispersed distributions. A bimodal distribution of aggregates and spheres is necessary. It is possible and physically

reasonable that the aggregates and spheres have their respective size distributions centered at our retrieved radii, but our results show that a combination of monodispersed aggregates and monodispersed spheres is sufficient for interpreting currently available observations.

**Test of refractive index**. As the temperature drops rapidly in Pluto's lower atmosphere from >100 K at 25–50 km to ~40 K near the surface[17], organic ice is expected to form[23], which may influence the scattering cross section at visible wavelengths. Laboratory measurements of $CH_4$ and $C_2H_4$ ices at 633 nm[58], which is similar to the wavelength of LORRI (608 nm), suggest that organic ices have typical real refractive indices near 1.5 at temperatures between 40–65 K. In comparison, the real refractive index of Titan "tholins", the haze material assumed in this work, is around 1.7. Also, due to the slightly higher CO mixing ratio in Pluto's atmosphere compared to Titan, Pluto's haze may contain more oxygen atoms than "tholins". Jovanović et al.[39] found through making laboratory analogues of Pluto's haze that, while the real refractive index is not very different from tholins, differences in the imaginary refractive index led to greater absorption in the visible and IR. These differences in the optical properties of haze particles influence the retrieved haze morphology. To address this issue, we conducted a sensitivity test by varying the haze refractive index (Supplementary Fig. 9). We change one of the real and imaginary parts of the refractive index and derive the corresponding changes in UV extinction cross section and visible scattering properties. For these tests we consider 1 μm aggregates with 20 nm monomers, the same as in our retrieval. Our results show that the scattering phase function is not sensitive to the change in refractive indices at all, which is due to the large particle size compared to the observation wavelengths. The scattering cross section is also largely insensitive to the imaginary refractive index, such that the greater absorption at visible and IR wavelengths due to the different haze compositions[39] should not influence the observed scattering quantities. A decrease in the real refractive index due to e.g. organic ice coatings results in both smaller extinction and scattering cross sections, as well as a smaller ratio of the scattering cross section in the visible to the extinction cross section in the UV, the quantities shown in Supplementary Fig. 8a, b. Therefore, the underestimation of backscattering in the visible is larger if "tholin" materials are substituted by $CH_4$ or $C_2H_4$ ice, which is likely true for other organic ices, reinforcing the need for a population of small spherical particles.

Although the uncertainty in the optical properties of Pluto's haze does not influence our interpretation of the bimodal distribution for haze morphology, the lack of laboratory measurements of Pluto haze analogs contributes to the difficulty in estimating Pluto's energy budget. If the haze were more absorbing at visible and IR wavelengths[39], how it influences heating and cooling rates in Pluto's atmosphere could differ from the previous estimates[28]. Moreover, ice condensation may introduce a completely different set of radiative transfer processes. As such, uncertainties in the haze composition and optical properties are now the bottlenecks for further improvements to models, with laboratory experiments that can measure both quantities being increasingly necessary if we want to further our understanding of Pluto's haze.

## Data availability

The New Horizons observations are available on NASA PDS (https://pds-smallbodies.astro.umd.edu/data_sb/missions/newhorizons/index.shtml). The measured haze optical properties are in Khare et al. [38]. The processed observations, including the extinction and scattering intensities, are attached in the Supplementary Information. The retrieved parameters describing haze morphology and corresponding scattering properties are also attached in the Supplementary Information. Source data are provided with this paper.

## Code availability

The data processing procedure is described step by step in the Methods. The Python package emcee for implementing MCMC is available at https://emcee.readthedocs.io. The haze scattering model is described in the appendix of Tomasko et al.[10]. The sphere pixelation tool is available at https://healpix.sourceforge.io.

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

## Acknowledgements
We thank William M. Grundy for sharing the LEISA data, Michael L. Wong and Xue Feng for improving figure representations, and Yan Wu for comments. P.G. is supported by NASA Hubble Fellowship grant HST-HF2-51456.001-A awarded by the Space Telescope Science Institute, which is operated by the Association of Universities for Research in Astronomy, Inc., for NASA, under contract NAS5-26555. X.Z. is supported by NASA Solar System Workings Grant 80NSSC19K0791. A.F.C. is supported by NASA under the New Horizons Project.

## Author contributions
S.F. conducted the data analysis, performed the calculations, and wrote the manuscript. S.F., P.G., X.Z., and Y.L.Y. conceived and designed the research. P.G. and D.J.A. provided the microphysical model. P.G. and C.L. provided the aggregate scattering model. X.Z. originated the idea of bi-modality. N.W.K. and C.J.B. contributed to the analysis of MVIC data. J.Y. contributed to interpreting and presenting the retrieval results. L.A.Y. and A.F.C provided insights into interpreting New Horizons observations. All authors contributed to the manuscript writing.

## Competing interests
The authors declare no competing interests.
