## [Peer Review File · Nature Communications]

Reviewers' Comments:

Reviewer #1:

Remarks to the Author:

Review report for NCOMMS-21-37784-T

Title: A bimodal distribution of haze in Pluto's atmosphere

The authors have made improvements to the manuscript and addressed most of the specific comments. Considering its important but incremental scientific advance, I still believe that this work would be more appropriate for a narrowly focused journal rather than Nature Communications.

Clarification is needed for Line 325-327: Does the authors mean that CO condenses on the haze particles? In addition, the authors need to define what haze particles are (Photochemically produced solids? Condensed ices? Or both?) and be consistent throughout the manuscript.

Reviewer #2:

Remarks to the Author:

The key result of this study is that the haze observations made by four instruments on New Horizons are best fit with a bimodal distribution of haze particles below 50 km altitude. This has important implications for haze formation processes and feedback from hazes in not only Pluto's, but also Titan's and Triton's atmospheres. I previously reviewed this manuscript for Nature Astronomy and am very pleased to see a revised version of the manuscript under consideration by Nature Communications.

The originality and significance of these results is much more apparent in the revised manuscript than the original and worthy of publication in this journal. In the original manuscript, the importance of a bimodal distribution was not clear. The authors have done a very good job in this revision at outlining the importance up front for the benefit of the type of audience reading Nature journals. The combined study of all four instrument observations is novel and important.

The data analysis is well done and the methodology is generally strong. Each of my concerns with the methodology in the previous version of the manuscript was sufficiently addressed in the revision. Sufficient detail is provided.

I recommend a few minor revisions:

Line 21 "encourage reanalysis of observations"

Lines 71-73 recommend revising to "their interactions with condensing/sticking gases and their role in atmospheric radiation, which is..."

Line 80 possibly revise to "the haze particles were thought to be fractal..."

Lines 122-124 Is there a reference that can be cited to justify the statement that temperatures will not drastically affect the major haze formation pathways? This seems like a very broad generalization that may not hold up when compared to laboratory studies of tholin formation.

Line 304 Hillier et al. 1991 does not propose water ice, but instead proposes "neutral ice condensates, probably N₂ ice."

Kathy Mandt

We thank the reviewers for the positive comments. The paper has been revised accordingly and further improved and clarified. In response, revisions have been made to accommodate the points raised by the reviewers and also to comply with journal format. Changes are marked bold in the manuscript.

Review report for NCOMMS-21-37784-T

Title: A bimodal distribution of haze in Pluto's atmosphere

Reviewer #1 (Remarks to the Author):

The authors have made improvements to the manuscript and addressed most of the specific comments. Considering its important but incremental scientific advance, I still believe that this work would be more appropriate for a narrowly focused journal rather than Nature Communications.

Clarification is needed for Line 325-327: Does the authors mean that CO condenses on the haze particles? In addition, the authors need to define what haze particles are (Photochemically produced solids? Condensed ices? Or both?) and be consistent throughout the manuscript.

We mean that CO condenses out of the atmosphere but less so than CH₄ so that the fraction of oxygen atoms in the photochemically produced haze particles increases. The sentence is revised to, “Also, when Pluto moves away from perihelion, the CO/CH₄ ratio is expected to increase, and therefore so does the fraction of oxygen atoms in the photochemically produced haze particles, as more CH₄ condenses out of the atmosphere onto the surface than CO when the temperature decreases.”

The haze particles are assumed to be photochemically produced solids, though we conducted a sensitivity test for the effects of possible ice content. To clarify, we revised the sentence to, “In the nominal forward model, we assume the haze particles are photochemically produced solids and use the complex refractive indices of ‘tholins’ ...”.

Reviewer #2 (Remarks to the Author):

The key result of this study is that the haze observations made by four instruments on New Horizons are best fit with a bimodal distribution of haze particles below 50 km altitude. This has important implications for haze formation processes and feedback from hazes in not only Pluto's, but also Titan's and Triton's atmospheres. I previously reviewed this manuscript for Nature Astronomy and am very pleased to see a revised version of the manuscript under consideration by Nature Communications.

The originality and significance of these results is much more apparent in the revised

manuscript than the original and worthy of publication in this journal. In the original manuscript, the importance of a bimodal distribution was not clear. The authors have done a very good job in this revision at outlining the importance up front for the benefit of the type of audience reading Nature journals. The combined study of all four instrument observations is novel and important.

The data analysis is well done and the methodology is generally strong. Each of my concerns with the methodology in the previous version of the manuscript was sufficiently addressed in the revision. Sufficient detail is provided.

I recommend a few minor revisions:

Line 21 "encourage reanalysis of observations"

Revised accordingly.

Lines 71-73 recommend revising to "their interactions with condensing/sticking gases and their role in atmospheric radiation, which is..."

Revised accordingly.

Line 80 possibly revise to "the haze particles were thought to be fractal..."

Revised accordingly.

Lines 122-124 Is there a reference that can be cited to justify the statement that temperatures will not drastically affect the major haze formation pathways? This seems like a very broad generalization that may not hold up when compared to laboratory studies of tholin formation.

We revised such a strong statement to, "Despite higher temperatures on Titan and lower CH₄ abundances on Triton, the major formation pathways of hazes are expected to be similar (Lavvas et al. 2010, Ohno et al. 2021)."

Two references to numerical simulations of Titan's and Triton's hazes are added. Lavvas et al. (2010) detailed the photochemical formation pathway of hazes on Titan; Ohno et al. (2021) found that haze particles can still grow to ~1 μm under the conditions of Triton's atmosphere and with much more gas condensation.

Line 304 Hillier et al. 1991 does not propose water ice, but instead proposes "neutral ice condensates, probably N₂ ice."

We revised this sentence as "... likely due to intense condensation of neutral ice, such as N₂".